# Voltage-Stabilizing Method of Permanent Magnet Generator for Agricultural Transport Vehicles

**Jianwei Ma [1,2,*], Liwei Shi [3] and Amir-Mohammad Golmohammadi [4,*]**

1   Department of Automotive Engineering, Hebei Vocational University of Technology and Engineering, Xingtai 054000, China
2   Hebei Special Vehicle Modification Technology Innovation Center, Xingtai 054000, China
3   School of Transportation and Vehicle Engineering, Shandong University of Technology, Zibo 255000, China
4   Department of Industrial Engineering, Arak University, Arak 38156879, Iran
*   Correspondence: kyc@xpc.edu.cn (J.M.); a-golmohammadi@araku.ac.ir (A.-M.G.);
    Tel.: +86-18931923929 (J.M.)

**Abstract:** Permanent magnet generators have the advantages of simple structure, high reliability, high efficiency, and energy saving. It is suitable for agricultural transportation vehicles, but there are some troubles on voltage regulation. In order to realize the stable output of permanent magnet generator, a kind of voltage-stabilizing method to ensure the average output voltage stability is proposed: by controlling the degree of clipping. First, the voltage regulation principle of permanent magnet generator is analyzed, mathematical model of permanent magnet generators in synchronous rotation coordinate system is built, and on this basis, the voltage-stabilizing circuit is designed. Second, the voltage-stabilizing circuit model of permanent magnet generator is created, the simulation analysis of reference point voltage and the output voltage under different speed and load is carried out, and the average value of output voltage is calculated according to the simulation curve taking advantage of the calculus principle. Third, the voltage-stabilizing circuit is made and tested. By comparing the simulation results with the experimental results, it is proved that the voltage-stabilizing circuit is suitable for the working characteristics of permanent magnet generator, the selected parameters of component are reasonable, and the simulation results are accurate and reliable. The circuit has excellent voltage-stabilizing performance. It provides a convenient and reliable method for the design and development of voltage-stabilizing circuit and promote the application of permanent magnet generator on agricultural transport vehicles.

**Keywords:** permanent magnet generator; voltage-stabilizing circuit; simulation analysis; agricultural transport vehicles

## 1. Introduction

The agricultural transport vehicles are a short-distance transport tool specially used for agriculture, circulation of agricultural products, and rural construction. Because agricultural transport vehicles can meet the needs of agriculture and forestry and meet the purchasing power of farmers, they have developed rapidly. At present, the number of agricultural vehicles exceeds 20 million [1–3]. Each agricultural transport vehicle needs to be equipped with a generator to provide power for the electrical equipment and charge the battery. Because the permanent magnet generator has the advantages of non-excitation winding, non-carbon brush slip ring and low power consumption, high efficiency and low failure rate, permanent magnet generator is suitable for agricultural transport vehicles. However, it is impossible to change the magnetic field strength by adjusting the excitation current as the silicon rectifier generator to achieve stable voltage control, which causes some difficulties to stabilize the output voltage. Therefore, one of the keys to the wide application of permanent magnet generator is to design a voltage-stabilizing controller that can achieve voltage stability in a wide speed and load range [4].

At present, the study on stabilized voltage control of permanent magnet generator is mainly divided into two categories: One is to directly adjust the generator to achieve stabilized voltage output using mechanical or electromagnetic mode; in the other class, the output voltage of permanent magnet generator is adjusted by power electronic device.

Lu. G. et al. proposed a method of adjusting the output voltage of the permanent magnet generator by adjusting the connection mode of the armature winding or the number of coil turns. When the rotor of the permanent magnet generator rotates at high speed, the coil is connected in parallel or the number of coil turns connected is reduced. When the rotor runs at low speed, the coil is connected in series or more coil turns are connected [5]. There are some disadvantages in the regulation control method, such as complex structure and low voltage regulation accuracy, it cannot meet the requirements of wide speed range and output voltage with small fluctuation of generator for agricultural transport vehicles.

Some scholars proposed optimization of a fuzzy automatic voltage controller for a generation system using real-time recurrent learning, which is a technique conventionally used for the training of recurrent neural networks [6,7]. It is not suitable for generator on agricultural transport vehicles.

For the permanent magnet generator used in extended range hybrid electric vehicle, the voltage stabilization problem directly affects the working condition prediction control and energy management strategy [8–10]. In [11,12], a new type of hybrid excitation claw pole machine (HE-CPM) for EV was proposed, 3D finite element analysis (FEA) and 3D magnet equivalent circuit were adopted.

For the axial permanent magnet generator, Javadi S et al. adopted the method of adjusting the relative distance between stator and rotor to stabilize the output voltage, the relative distance is increased at high speed, and the relative distance is reduced at low speed [13].

Capponi, F.G. et al. segmented the permanent magnet of rotor, the effective magnetic flux reaches maximum when the phases of the permanent magnets are the same, the effective excitation magnetic field was adjusted by controlling the angle of permanent magnets, so as to control output voltage stability [14,15].

Tian, X.P. et al. introduced three-phase alternating current into the electric excitation winding to generate a rotating synchronous magnetic field with the rotor. The electric excitation magnetic field adjusted the integrated air gap magnetic field to control the output voltage. In this way, the electric magnetic field and permanent magnetic field are coupled with each other, and the excitation current needs to be controlled according to the rotor position signal, which is difficult to design and complicated to control [16].

In short, the effective magnetic field of permanent magnet generator can be adjusted directly or indirectly through internal adjustment methods, but the increase of magnetic field adjustment mechanism will make the mechanism of permanent magnet generator complex, so it is not suitable for permanent magnet generator in agricultural transportation vehicles.

Zhang, X.Y. and Chou, N.C. et al. used silicon-controlled rectifier to realize the regulated output of permanent magnet generator, and designed single-phase, three-phase, and five phase controllable rectifier voltage-stabilizing devices [17–19], which realized the stability of output voltage and ensured high efficiency, but this method cannot be applied to the voltage stabilization of high-power generator.

For the six-phase alternating current generated by the six-phase permanent magnet synchronous generator, Wang, Y.L. et al. adopted a switching regulated rectifier circuit with a dual-three-phase fully controlled bridge parallel to rectify the six-phase alternating current. Closed-loop control was adopted to output constant direct current which did not change with the changes in speed and load. Using MATLAB\Simulink and PSB, the rectifier circuit simulation model of six-phase permanent magnet synchronous generator is built. The simulation is carried out [20,21]. However, the circuit with precise parameters was not designed according to the actual requirements, and the principle was only verified by simulation, and regulating circuit is not verified by experiment.

The main aim of this paper is to design a new voltage-stabilizing circuit which is suitable for permanent magnet generators. The content of the paper is arranged as follows: in Section 2, advantages of permanent magnet generator are analyzed. The reason why permanent magnet generator is suitable for agricultural transport vehicles application is expounded. In Section 3, mathematical model in synchronous rotation coordinate system is built. In Section 4, voltage-stabilizing principle is analyzed. In Section 5, a new voltage-stabilizing circuit which is suitable for permanent magnet generators is designed. In Section 6, voltage-stabilizing circuit model is created and simulation of voltage regulation performance is carried out. In Section 7, According to the simulation waveform, the theoretical average voltage is calculated. In Section 8, by comparing the experimental data with the simulation results, the feasibility of the simulation method and the reliability of the simulation results are verified. Thus, a voltage-stabilizing control circuit suitable for permanent magnet generator with excellent voltage-stabilizing performance is designed. Finally, some important conclusions are given. In a word, the mathematical model of permanent magnet generator is established, and its voltage-stabilizing principle is analyzed according to the mathematical model. By combining theoretical analysis with simulation calculation, the voltage-stabilizing control circuit of permanent magnet generator which is suitable for agricultural transport vehicles is designed. The experiment shows that the voltage-stabilizing control circuit has excellent voltage regulation effect. The structure of the circuit is simple and the cost is low, which can promote the permanent magnet generator to be widely used in the agricultural transport vehicles and improve the whole performance of the electrical system of the agricultural transport vehicles. There is guiding significance to the design and development of the voltage-stabilizing circuit which is suitable for permanent magnet generator of agricultural transport vehicles.

## 2. Analysis of Advantages of Permanent Magnet Generator

### 2.1. Simple Structure and High Reliability

The permanent magnet generator adopts Nd-Fe-B permanent magnet, it is non-carbon brush slip ring structure and non-electric excitation winding [22]. Its overall structure is simple, which avoids the problems of easy burning, wire breaking, wear of carbon brush and slip ring, and the reliability is improved [23,24].

### 2.2. High Efficiency and Low Consumption of Electricity

The generator excites using permanent magnet, does not need excitation winding, and consume electrical energy, which improves the efficiency. The permanent magnet rotor can provide rotating magnetic field without excitation current. There is no mechanical loss between carbon brush and slip ring of electrically excited rotor, and the efficiency can be improved by more than 5%. It can be seen from Figure 1. The efficiency of silicon rectifier generator is 15–25% lower than that of permanent magnet generator [25,26].

### 2.3. Power Supply Performance Is Excellent at Low Speed, Output Voltage Adjustment Rate Is Low

When speed of generator is low, under condition of the same electromagnetic parameters, when the excitation current of silicon rectifier generator is 0, the permanent magnet generator can output 3 A–5 A, and the excitation current is saved, which will greatly improve the power supply performance at low speed. The structure of rotor is simple and processing technic is convenient. Magnetic leakage and armature reaction reactance of direct axis and quadrature axis are little. Therefore, the voltage regulation rate is little, its external characteristics curve is relatively flat, and it has excellent output voltage stability under high-speed power [27].

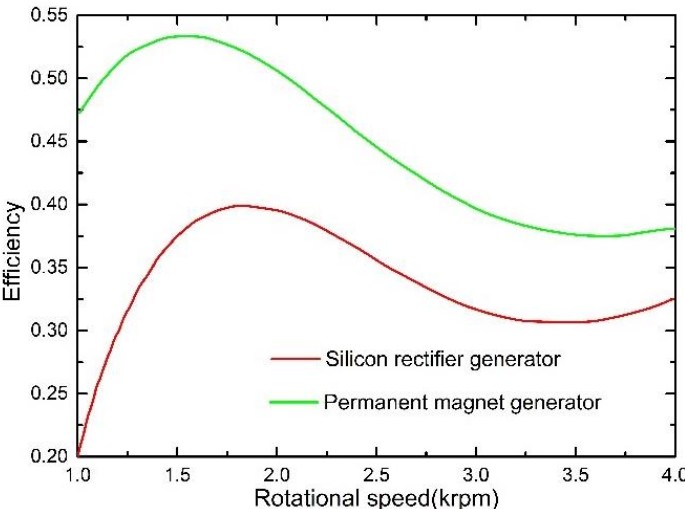

**Figure 1.** Comparison of efficiency characteristic curves between permanent magnet generator and silicon rectifier generator.

### 2.4. Excellent Environmental Adaptability

In the generator, there are no excitation winding, carbon brush, and slip ring, which can operate in the harsh conditions of humidity or dust. Without external excitation source, the generator can generate electricity as long as it rotates, and the reliability is improved. The electromagnetic interference caused by the friction between the carbon brush and the slip ring is eliminated and the electric spark is eliminated. It can also be well applied in high explosive and dangerous environments [28]. At the same time, the permanent magnet generator can be well used in nuclear power generation, and the circuit can be used in the field of novel nuclear systems of IV generation [29].

### 2.5. Extended the Service Life of Battery

First of all, charging performance of the permanent magnet generator is very good at low speed, which can make the battery always in a sufficient state, and the vulcanization of the battery plate can be effectively prevented. Second, the voltage-stabilizing precision is high, there is no under charge or overcharge, the trace gas state is maintained during the charging process, and a large number of bubbles will not be generated, which will neither produce a large amount of electrolyte loss nor pollute the battery surface. The loss of active substances caused by serious exhaust can be effectively avoided, and the service life of the battery can be improved. Third, the half wave controllable rectifier regulator circuit is essentially a chopper switching regulator. During operation, part of the wave is cut off. At the time of cutting off, the charging stops and the battery polarization dissolves rapidly. If the load is connected, the battery will discharge, which is equivalent to reverse charging. It can produce a better effect of eliminating polarization and provide good conditions for charging at the conduction time. At the conduction time, high amplitude pulse charging current can make the active substances on the electrode plate conduct electrochemical reaction fully. Therefore, it can increase the battery capacity and prolong the service life of the battery [30].

### 3. Mathematical Model Building in Synchronous Rotation Coordinate System

In this paper, according to the generator convention, the positive direction of each parameter is specified. It is assumed that the counterclockwise rotation direction of the rotor is the positive rotation direction, the positive directions of three-phase winding flux chain $\psi_a$, $\psi_b$, $\psi_c$ are consistent with the positive directions of a, b, and c three axis, the polarity of the terminal voltage of the three-phase winding and the positive direction of the phase current are defined in accordance with the convention of the generator, that is, the $i_a$ flows out of the generator from the positive pole of the terminal voltage, the positive

direction of $T_e$ is opposite to the positive direction of the speed, and the positive direction of the mechanical torque $T_L$ input from the shaft is consistent with positive direction of the speed [31]. The conventional positive direction of permanent magnet generator is shown in Figure 2. According to the above assumptions, the structure model of permanent magnet synchronous generator is established, it is shown in Figure 3.

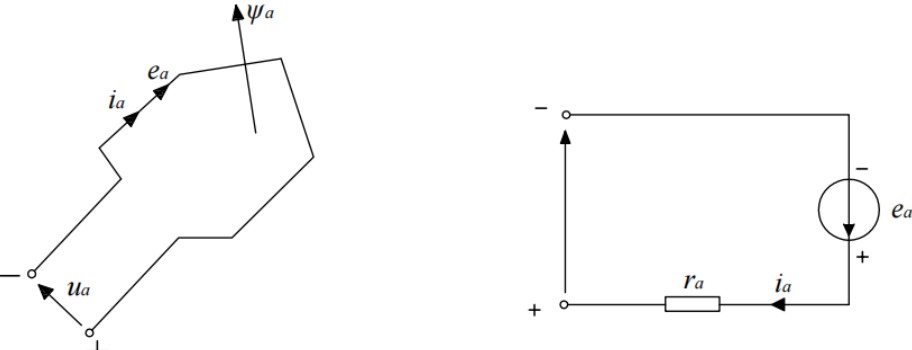

**Figure 2.** Conventional positive direction of permanent magnet generator.

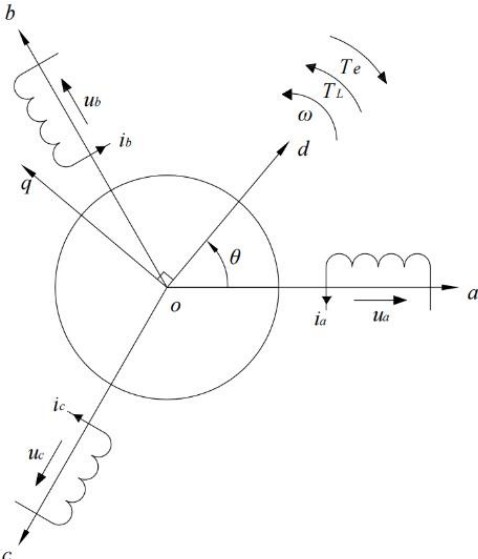

**Figure 3.** Structure model of permanent magnet generator.

In order to study the permanent magnet generator system, the mathematical model of the permanent magnet generator in the synchronous rotation orthogonal coordinate system is established. Constant phase amplitude transformation is adopted in this paper. Since the neutral point of the star network three-phase motor is generally suspended and the zero-sequence current is zero, the 0-axis component is not considered in the generator system, and the corresponding coordinate transformation matrix is shown in Formula (1).

$$G_{3/2r} = \frac{2}{3} \begin{pmatrix} cos\theta & cos\left(\theta - \frac{2\pi}{3}\right) & cos\left(\theta + \frac{2\pi}{3}\right) \\ -sin\theta & -sin\left(\theta - \frac{2\pi}{3}\right) & -sin\left[\theta + \frac{2\pi}{3}\right] \end{pmatrix} \tag{1}$$

The three-phase winding adopts the generator convention, positive current produces negative flux, and the mathematical model of permanent magnet generator in $d$-$q$ coordinate system can be obtained by constant phase amplitude transformation. Flux equation, voltage equation, torque equation, and motion equation are shown in Formulas (2)–(5) respectively [32].

$$\begin{cases} \psi_d = -L_d i_d + \psi_f \\ \psi_q = -L_q i_q \end{cases} \tag{2}$$

$$\begin{cases} u_d = -R_s i_d - L_d \frac{di_d}{dt} + \omega L_q i_q \\ u_q = -R_s i_q - L_q \frac{di_d}{dt} + \omega \psi f - \omega L_d i_d \end{cases} \tag{3}$$

$$T_e = \frac{3}{2} p [\psi f + (L_q - L_d) i_d] i_q \tag{4}$$

$$T_L - T_e = \frac{B\omega}{p} + J \frac{d\Omega}{dt} \tag{5}$$

In Formula (2)–(5), $\psi_D$ and $\psi_Q$ are the direct and cross-axis components of stator flux linkage, $L_d$ and $L_q$ are the synchronous inductance of direct and cross-axis, $\psi_F$ is the amplitude of fundamental wave component of excitation flux linkage of permanent magnet, $\omega$ is the electric angular velocity of permanent magnet generator, $R_s$ is the resistance of each phase of stator windings, p is the polar logarithm, $T_e$ is the electromagnetic torque, $T_L$ is the mechanical torque input from the shaft of the generator, $\Omega$ is the mechanical angular velocity of the generator, and $B$ is the friction coefficient of the generator.

## 4. Analysis of Voltage-Stabilizing Principle

The external load of permanent magnet generator for agricultural transport vehicles mainly includes the light, which can be considered as pure resistance and resistance value is constant. When the external circuit is switched on, the current through the load is shown in Formula (6).

$$I = \frac{4k_B f N k_w \Phi_0 \times 10^{-8}}{\sqrt{(R + r_0)^2 + x_d^2}} \tag{6}$$

where, $k_B$ is the waveform coefficient, when the magnetic field in gap is sinusoidal, $k_B = 1.11$; $f$ is the frequency, $f = pn/60$, $p$ is the polar logarithm, it is a synchronous generator with 12 poles in the design, $p = 6$, $n$ is the speed, its unit is r/min; $N$ is the number of turns of per phase armature winding; $K_w$ is the motor winding coefficient, $K_w = 0.92$; $\Phi_0$ is the effective magnetic flux through the armature winding, $\Phi_0 = \Phi_\delta / \sigma_0$, $\sigma_0$ is the flux leakage coefficient, in star rotor with polar shoe, $\sigma_0 = 1.2–1.5$; $\varphi_\delta$ is the magnetic flux of each pole in the gap, $\Phi_\delta = \alpha B_\delta \tau L \delta$, $\alpha$ is the polar arc coefficient, in star rotor with polar shoe, $\alpha = 0.75–0.85$; $B_\delta$ is the no-load magnetic induction intensity, $B_\delta = (0.75–0.85) B_\gamma$, $B_\gamma$ is the remanence magnetic induction intensity; $\tau$ is the polar distance, $\tau = \pi D/(2p)$, and $D$ is the inner diameter of stator core. $L_\delta$ is axial calculated length of air gap; so, $E = 6.81 \times 10^{-10} p \cdot N \Phi_0 \cdot n = C_e n$, $C_e$ is a constant. $R$ is the resistance of the light; $r_0$ is the resistance of armature winding, $r_0 = l \rho/s$, $l$ is the length of winding coil, $\rho$ is the resistivity of conductor, its unit is $\Omega \cdot$m, $s$ is the cross-sectional area of winding coil; $X_d$ is the synchronous reactance of the armature winding, $X_d = X_S + X_a$, $X_S$ is the leakage reactance of the winding, which is shown in Formula (7); $X_a$ is armature reaction reactance, which is shown in Formula (8).

$$X_s = 15.5 \frac{f}{100} \cdot \left(\frac{N}{100}\right) \frac{L_1}{p} \sum \lambda \times 10^{-2} = 2.58 \times 10^{-9} N^2 L_1 \sum \lambda \cdot n = C_s \cdot n \tag{7}$$

$$x_a = \frac{2EF_a k_{aq}}{I \cdot \sum F} = \frac{6.13 \times 10^{-10} m k_{aq} N^2 \Phi_0}{\sum F} \cdot n = C_a \cdot n \tag{8}$$

where, $L_1$ is the length of stator core, $\Sigma\lambda$ is the total magnetic conductivity, $F_a$ is armature magnetomotive force of per pole, $F_a = 0.45$ m$Nk_{dp}I/p$, m is phase number, $K_{dp}$ is winding factor, $K_{dp} = 1$; $\Sigma F$ is the total magnetic potential difference; $K_{aq}$ is the conversion coefficient of magnetomotive force of cross axis armature. It is shown in Formula (9). $C_s$, $C_a$, and $C_d$ are all constants.

$$k_{aq} = \frac{\alpha\pi - sin\alpha\pi + 2cos(\alpha\pi/2)/3}{4sin(\alpha\pi/2)} \tag{9}$$

$$x_d = x_s + x_a = (C_s + C_a) \cdot n = C_d \cdot n \tag{10}$$

The load terminal voltage is shown in Formula (11).

$$U = IR = \frac{E}{\sqrt{(R + r_0) + x_d}} \cdot R = \frac{C_e n}{\sqrt{(R + r_0)^2 + (C_d n)^2}} \cdot R \qquad (11)$$

When the speed is very low, $(C_d n)^2$ is much less than $(R + r_0)^2$ and can be omitted, $U = C_e n \, R/(R + r_0)$, $U$ is proportional to $n$. When $n$ is very high, $(C_d n)^2$ is much larger than $(R + r_0)^2$, $(R + r_0)^2$ can be ignored, and $U = C_e R/C_d$, which is close to a constant.

It can be seen that when the rotational speed is low, the output voltage of the generator increases proportionally with the increase of the rotational speed, and then slows down with the increase of the rotational speed and becomes stable at the highest rotational speed. Therefore, under the condition of constant load $R$, the generator itself has a certain self-regulating voltage function. But in practical application, due to the change in the speed of the engine, load $R$ also often changes, so the output voltage of generator changes, ensuring the reliability and safety of power facilities. At the same time, considering the cost of generators for agricultural transport vehicles, the single-phase half-controlled bridge rectifier voltage stabilization method is adopted to stabilize the output voltage at high speed.

## 5. The Design of the Voltage-Stabilizing Circuit

### 5.1. Determination Parameters of Main Components in Circuit

#### 5.1.1. Determination of Output Power of Generator

In order to facilitate calculation and analysis, the concept of electrical frequency coefficient is introduced [33]. According to different seasons and environments, the usage opportunity of various electrical appliances is also different. In other words, the frequency coefficient of electrical appliances is also different. Taking generator of the model of 7YP-1150DA28-1 three-wheel agricultural vehicles manufactured in Rizhao city, Shandong Province, China (Wuzheng group Co., Ltd.) for example, the generator provides DC power for lighting, steering command, horn, wiper, and heater. The power of light is 100 W, the electrical frequency coefficient is 1.0, the power of starter is 1200 W, the electrical frequency coefficient is 0.1, the power of brake light is 20 W, the electrical frequency coefficient is 0.75, the power of turning light is 20 W, the electrical frequency coefficient is 0.1, the power of wiper is 60 W, and the electrical frequency coefficient is 0.25, the power of heater is 60 W, and the electrical frequency coefficient is 0.25.

According to Formula (12), the output power of the generator is no less than 267 W.

$$P = 100 \times 1.0 + 1200 \times 0.1 + 20 \times 0.75 + 20 \times 0.1 + 60 \times 0.25 + 60 \times 0.25 = 267\text{W} \quad (12)$$

According to the electric facilities of the three-wheel agricultural vehicles, the total amount of electricity is about 270 W, so the output power designed is 300 W.

#### 5.1.2. Determination of Speed Range

The rated speed of the generator is generally from 2000 r/min to 4800 r/min, and the rated speed of the generator selected is 4000 r/min in the design.

#### 5.1.3. The Determination of the Stable Voltage Range

The rated voltage of the generator for agricultural transport vehicles is 14 V. According to the requirements of the machinery industry standard GB/T23903-2009 of the People's Republic of China, the generator speed is between 2000 r/min and 4800 r/min, and its output voltage is between 13 V and 14.5 V. In the design, when the rotating speed of permanent magnet generator is from 2000 r/min to 4800 r/min, its stable voltage range is selected between 14 V and 14.5 V.

### 5.1.4. Main Electrical Components Parameters Calculation

Under the condition that the DC voltage anti-interference performance is satisfied, the capacitance calculation formula can be known from Formula (13).

$$C > \frac{1}{2\Delta V_m * R_{dc}} \tag{13}$$

Among them, the percentage of maximum voltage dynamic drop is generally about 6%, so the value of $\Delta V_m$ is selected as 6%. Resistance of rated load, $R_{dc} = \frac{U}{n} = 3.52$. Since the capacity of the capacitor is always larger than the value calculated, the capacity of the capacitor is 9 uF in preliminary design.

According to requirements of the circuit, model of thyristor is selected as NB60.

### 5.2. Design of the Voltage-Stabilizing Circuit

The rectifier circuit composed of two thyristors is a part of the voltage-stabilizing circuit. It replaces the full wave bridge rectifier circuit composed of four rectifiers, which reduces power consumption, improves output power, and realizes stable DC output. Since the voltage of on-board generator is only 14 V, but the power is large and the output current is also large, the current of each PN junction decreases by 0.7 V. According to the Equation q = UIT, the greater the current is, the greater the heat is. The large amount of heat generated by PN junction not only wastes the effective power of the generator, but also causes heat dissipation. When the output power of the generator is the same as that of full wave bridge rectifier circuit, the power consumption of the voltage-stabilizing circuit is lower than that of the full wave bridge rectifier circuit, which improves the output power of the generator. Therefore, the generator with rectifier voltage-stabilizing method has the advantages of simple structure, reliable performance, and low cost. The schematic diagram of voltage-stabilizing circuit is shown in Figure 4.

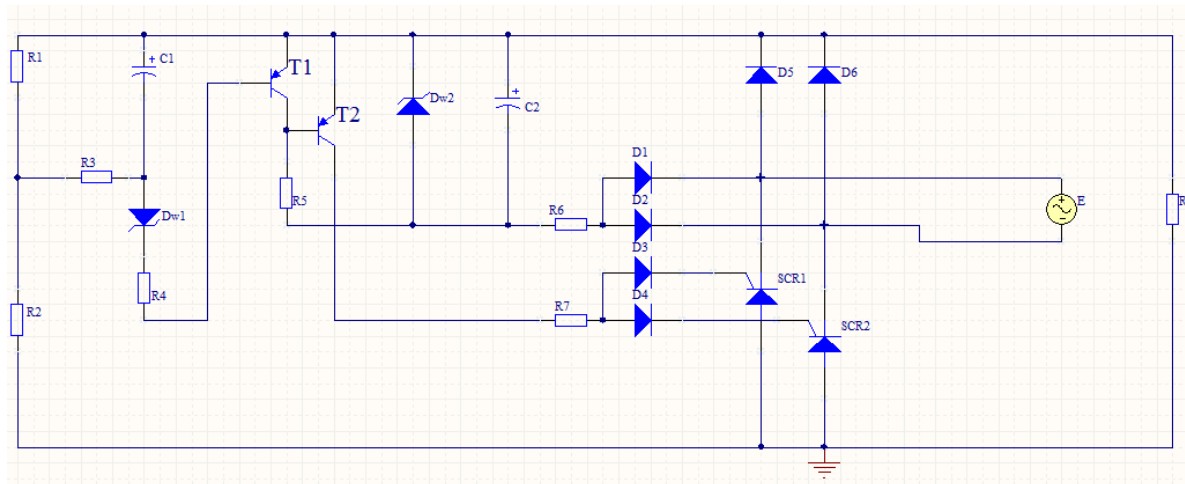

**Figure 4.** Schematic of voltage-stabilizing circuit. RL is rheostat; R1, R2, R3, R4, R5, R6, R7 are the resistance; C1, C2 are the capacitance; D1, D2, D3, D4 are the diodes; D5, D6 are the rectifiers; Dw1, Dw2 are stabile tubes; T1, T2 are triodes; SCR1, SCR2 are silicon controlled rectifiers.

By adjusting the resistance value of rheostat RL, the voltage of reference point A can be changed, and the target voltage-stabilizing value of the voltage-stabilizing circuit can be set to 14 V, U0 = 14 V.

When the generator runs at low speed, it mainly relies on Nd-Fe-B permanent magnet material with high remanence intensity to generate magnetic field. The structure design of the generator is improved, the number of poles is increased, the number of armature windings is increased, and the high output voltage is guaranteed.

When the generator starts to rotate, because speed is low, the output voltage U is also low, which is less than the target voltage-stabilizing value U0. The voltage between emitter of the triode T1 and point A is less than U1 which is breakdown voltage of the stabile tube Dw1, triode T1 is on the cut-off state. The voltage between the emitter and the base of the triode T2 is greater than 0.7 V, so the triode T2 conducts, and the current of collector goes through the resistor R7, diode D3, diode D4, and supplies gates of thyristors SCR1 and SCR2 with the trigger current respectively, makes controlled silicon to conduct, then DC is output [34].

The speed of generator further increases, the output voltage U increases, and the emitter voltage of the triode T1 and point A also increases. When the output voltage U is greater than the set target voltage-stabilizing value U0, the potential difference between the emitter of triode T1 and point A is greater than the breakdown voltage U1 of the stabile tube Dw1, and the triode T1 changes from the cut-off state to the conduction state. After the triode T1 is turned on, the voltage between the emitter and the collector is from 0.2 V to 0.3 V, which is less than the threshold voltage 0.7 V between the emitter and the base of the triode T2. The state of Triode T2 changes from on to off, and the gates of thyristors SCR1 and SCR2 no longer provide trigger current. Until there is no positive voltage, the output voltage U drops rapidly, and the voltage of triode T1 emitter and point A also drops, SCR1 and SCR2 will not turn off. When the output voltage U is lower than the set target voltage-stabilizing value U0, the triode T1 is disconnected, T2 is connected, and the thyristor rectifier is connected again to output DC. When the output voltage U rises again (greater than the set target voltage-stabilizing value U0), the regulator tube Dw1 is broken down again, the triode T1 is on, T2 is closed again. Triode T1 and T2 are repeatedly in the switching state. Through chopper and rectification, the generator output voltage is guaranteed to be stable DC.

## 6. Modeling and Simulation of Voltage-Stabilizing Circuit

### 6.1. Modeling of Voltage-Stabilizing Circuit

A new schematic file "*. Schdoc" is created, open the editing environment and enter the schematic editing environment; click the menu command "Design AD/move libraries" to enter the component library [35], add the components shown in Figure 1, call the "Edit align" command, and place the components in the corresponding position of the drawing; set the parameters of components and set the simulation properties of all components (assuming that all components are ideal components); conduct electrical connection with wires and ERC check for the whole circuit to ensure that there is no error in the whole circuit. Set the voltage frequency generated by the generator to 400 Hz and the peak voltage to 48.96 V (measured data when the generator is tested at the speed of 4000 r/min). Simulate the voltage value generated by the generator at different speeds by changing the frequency and peak value. Set the network node "VIN" at the output end of the voltage-stabilizing circuit. The output effect of the voltage-stabilizing circuit is obtained by observing the voltage waveform of the point.

### 6.2. Simulation Analysis

6.2.1. Simulation Analysis of Reference Point Voltage

The voltage of the reference point A is determined by the value of capacitance C1, the volatility of the reference point voltage can directly affect the switching frequency of the transistor, and the switch of the transistor controls the voltage-stabilizing output of the circuit. When the voltage of the reference point A is relatively stable, the voltage-stabilizing characteristics of the circuit can be improved. Therefore, whether the value of C1 is appropriate is crucial. Then, the value of capacitor C1 is changed, when the value of capacitor C1 is set as 0.1 μF, 0.9 μF, and 2 μF respectively, voltage signal of point A is moved to the "Active Signals list". The voltage signals of point A are displayed as results in the simulation waveform window, the voltage-stabilizing circuit is simulated and the voltage waveforms of the reference point A are shown in the Figures 5–7.

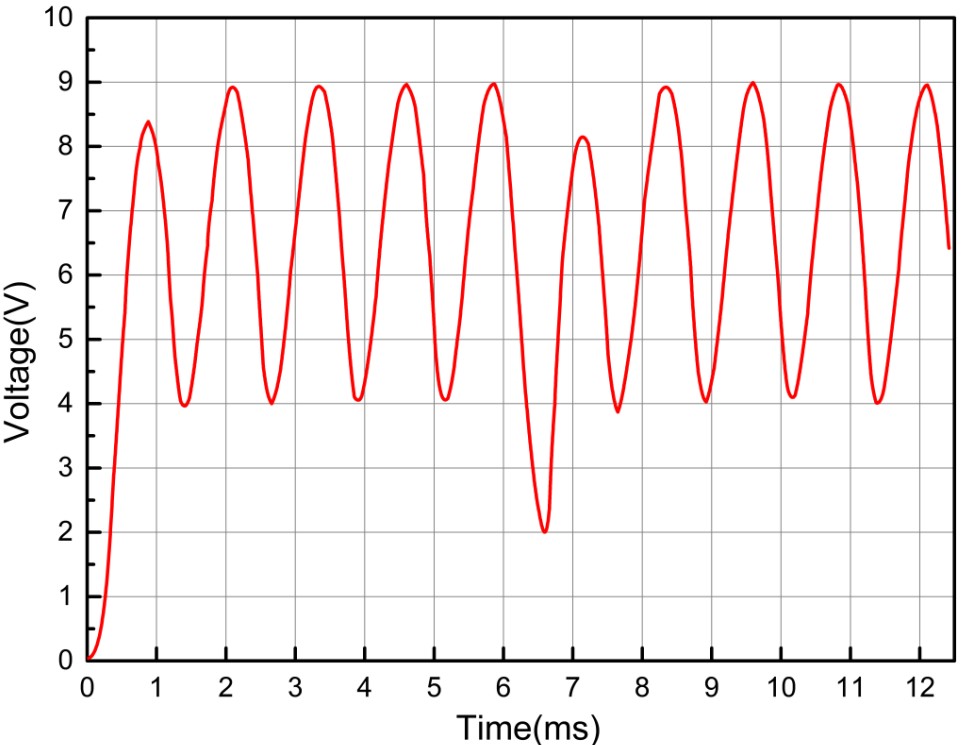

**Figure 5.** The voltage waveform of reference point A when value of C1 is 0.1 μF.

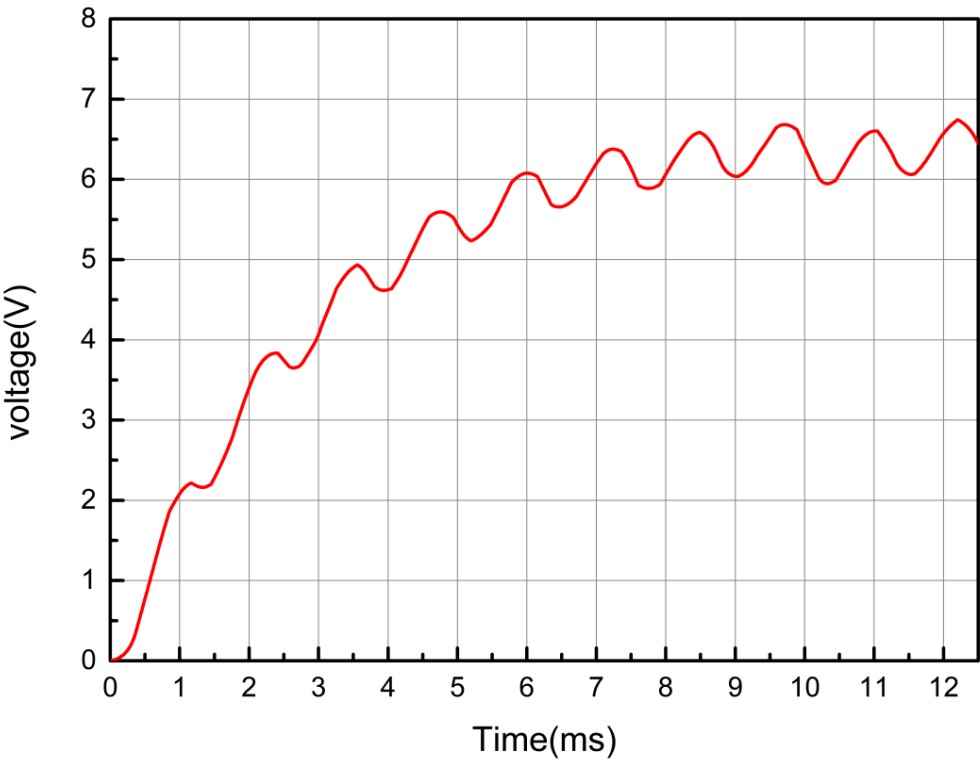

**Figure 6.** The voltage waveform of reference point A when value of C1 is 0.9 μF.

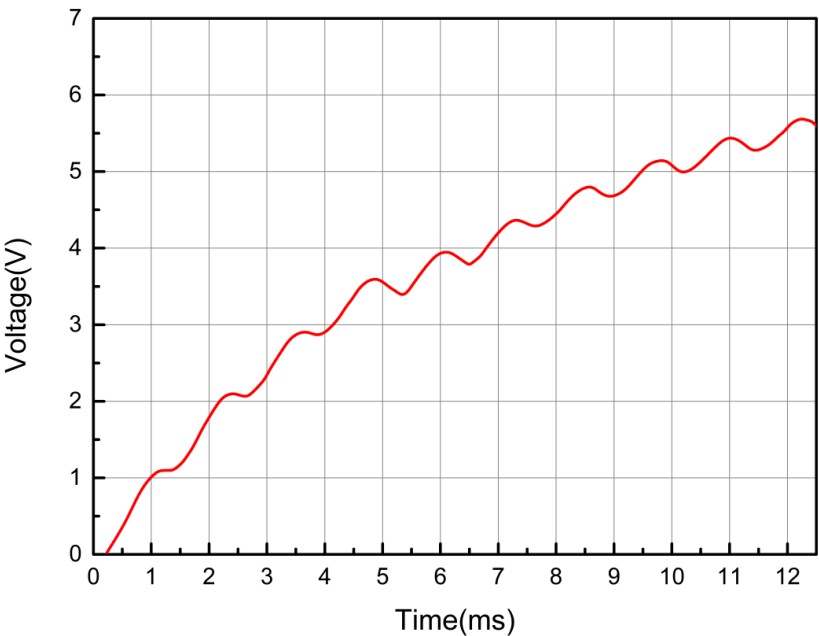

**Figure 7.** The voltage waveform of reference point A when value of C1 is 2.0 μF.

According to the waveform analysis of the analog output results in Figures 5–7, it can be concluded that when the value of capacitor C1 is 0.9 μF, the output voltage waveform of reference point A is more stable, so the value of C1 is selected as 0.9 μF.

### 6.2.2. Simulation Analysis of Output Voltage

After all parameters are set, select the menu "design\simplification\simulation" in the schematic editing window to start the simulation function and pop up the simulation analysis setting dialog box, so as to obtain the voltage waveform after stabilization, and select "working point analysis" and "transient/Fourier analysis". Set the start time, stop time, and step time interval of simulation; then, the VIN signal in the "available signal" is selected as the output signal, and click the "OK" button to start the simulation. Change the frequency and amplitude of the AC voltage source and repeat the above process to obtain the voltage waveform of the generator under different output voltages and different load conditions. The simulation results are shown in Figures 8–11.

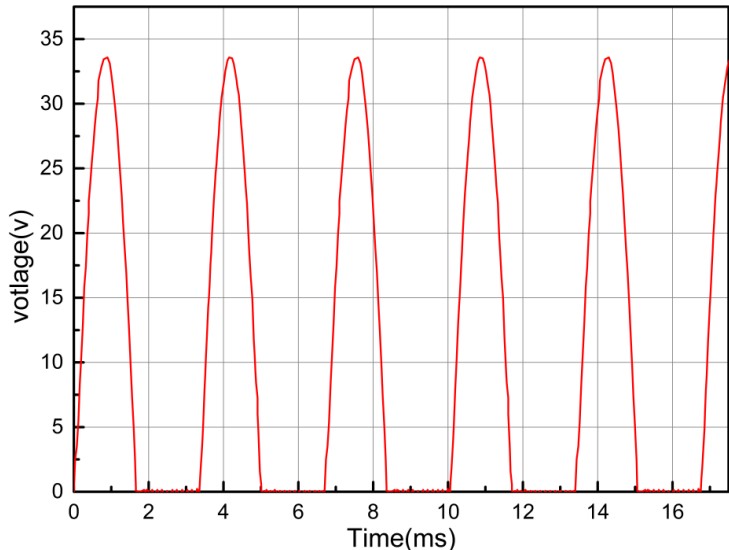

**Figure 8.** Output voltage waveform at 3000 r/min and no-load.

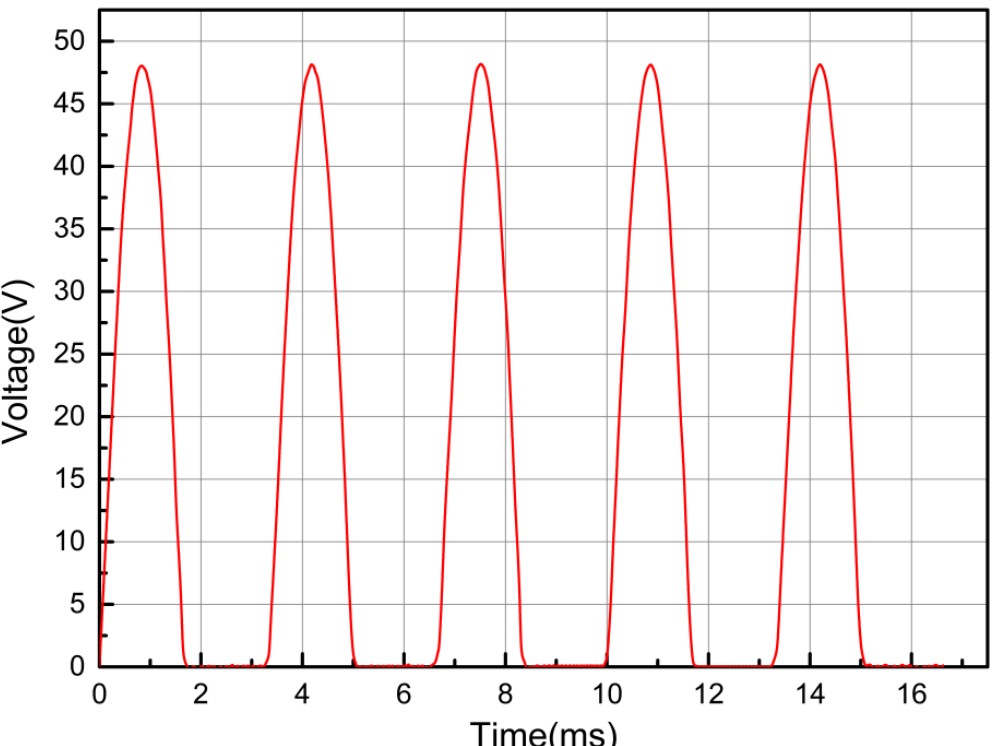

**Figure 9.** Output voltage waveform at 4000 r/min and no-load.

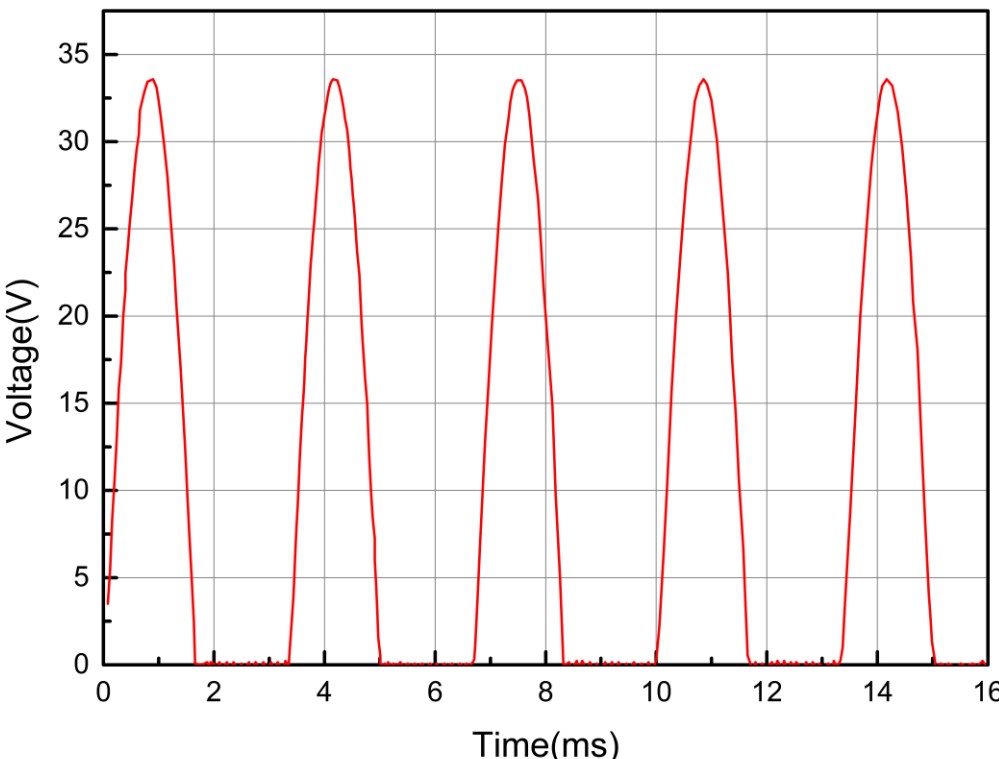

**Figure 10.** Output voltage waveform at 3000 r/min and 300 W.

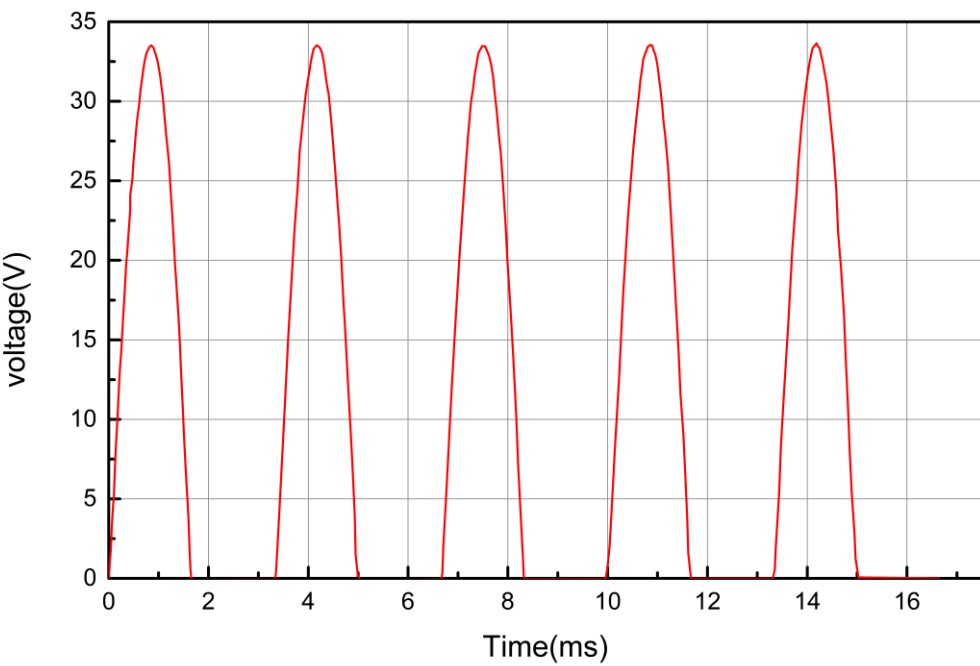

**Figure 11.** Output voltage waveform at 4000 r/min and 300 W.

6.2.3. Analysis and Calculation of Simulation Results

According to the simulation waveform of the output voltage, combining with the principle of calculus, the average value of the output voltage can be calculated. The calculation formula of output voltage is Formula (14).

$$\overline{U} = \frac{1}{T} \int_0^t E_m Sin2\pi f t dt \tag{14}$$

where, $\overline{U}$ is average voltage, $T$ is period, $E_m$ is peak voltage, $f$ is frequency. $p = \frac{np}{60}$, $p$ is pole pair logarithmic, $n$ is speed, the value of $p$ is 8 in the generator.

Taking the 4000 r/min and the 300 W as an example, the simulation result is substituted into Formula (14), and the average value of output voltage is calculated to be 14.267472 V at that moment; the calculation process is as follows. Similarly, the average voltage in other cases can be calculated. Simulation values of average output voltage under different speeds and loads are shown in Table 1.

$$U = \frac{1}{T} \int_0^t E_m Sin2\pi f t dt = \frac{1}{0.0025} \int_0^{0.0012} Sin2\pi \times 400t dt = 14.267472 \tag{15}$$

**Table 1.** Simulation calculation results of the output voltage.

| Speed (r/min) | 2000 | | | 3000 | | | 4000 | | |
|---|---|---|---|---|---|---|---|---|---|
| Load (W) | 60 | 150 | 300 | 60 | 150 | 300 | 60 | 150 | 300 |
| Voltage (V) | 14.12 | 14.11 | 14.01 | 14.30 | 14.24 | 14.21 | 14.36 | 14.45 | 14.27 |

## 7. Performance Experiment

Based on the above simulation results and the designed PCB board, a voltage-stabilizing controller is made and is encapsulated in an aluminum housing, then it is installed on a 12-pole radial excitation Nd-Fe-B permanent magnet generator. The voltage-stabilizing controller was installed on the permanent magnet generator prototype, and are installed on the model LDF-3 automotive electrical experiment platform to simulate various working

conditions of the agricultural transport vehicles; it is shown in Figure 12. Under the conditions of 60 W, 150 W, and 300 W respectively, the performance experiment is executed from low speed to high speed. Actual measurement results are shown in Table 2. The comparison between simulation results and experiment results is shown in the Figure 13.

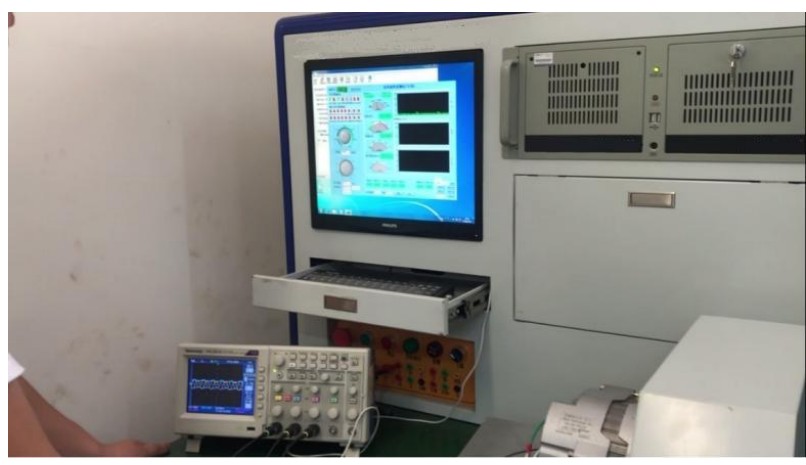

**Figure 12.** The experimental scene.

**Table 2.** Experiment results of output voltage.

| Speed (r/min) | 2000 | | | 3000 | | | 4000 | | |
|---|---|---|---|---|---|---|---|---|---|
| Load (W) | 60 | 150 | 300 | 60 | 150 | 300 | 60 | 150 | 300 |
| Voltage (V) | 14.01 | 14.04 | 13.82 | 14.20 | 14.15 | 14.12 | 14.40 | 14.28 | 14.18 |

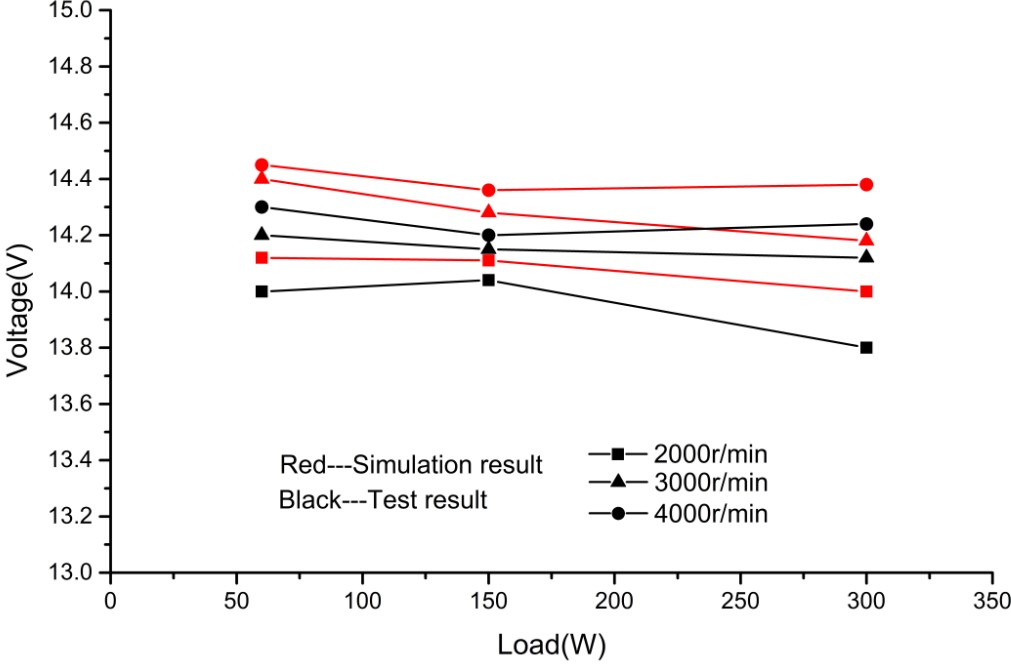

**Figure 13.** Comparison between simulation results and experiment results.

From Table 2 and Figure 13, it can be seen that the simulation results are close to the experiment results, the maximum error is 1.4% and the experimental data are slightly larger than the simulation data, the reason is that the components in simulation model are considered as ideal components, and the power loss on the components is little. However, experiment results meet the standards of the People's Republic of China Machinery Industry GB/T23903-2009 [36]. It is verified that the principle of the voltage-stabilizing circuit is

feasible, and the parameters of the electronic components determined by simulation are reasonable. By comparing with the experimental data, it is proved that the simulation method is correct and the simulation results are reliable. Therefore, the voltage-stabilizing circuit designed according to the simulation results has excellent voltage-stabilizing performance.

According to the mathematical model of permanent magnet generator in synchronous rotation coordinate system and the analysis of voltage-stabilizing control principle, the voltage-stabilizing control circuit is designed, and the simulation analysis is carried out; and the voltage-stabilizing control circuit is made. After the experimental analysis, it shows that the voltage-stabilizing control circuit has excellent effect and is composed of simple electronic components, can be well applied to the agricultural transport vehicle, the circuit can promote the application of permanent magnet generator, improve farm transporter of the stability of power system, increase the service life of the battery, reduce the failure rate of the generator.

## 8. Conclusions

In order to solve the problem of voltage-stabilizing output of permanent magnet generator for agricultural transport vehicles, according to the output characteristics of permanent magnet generator and the operating conditions of agricultural transport vehicles, this paper mainly studies the voltage-stabilizing control method of permanent magnet generator for agricultural transport vehicle. According to the performance index of the generator for agricultural transport vehicles, the component parameters of the voltage-stabilizing circuit are calculated, and the voltage-stabilizing control circuit is designed. The simulation model of voltage-stabilizing circuit is created. The influence of capacitor C1 on reference voltage and the output voltage under different load and different speed is simulated and analyzed. In addition, the average value of the output voltage is calculated taking advantage of the integral theory, and the accurate results are obtained. According to circuit designed, a voltage-stabilizing controller is made and it is installed on the permanent magnet generator. The performance experiments are carried out. In the future, the study on reliability of the circuit and manufacturing technique should be carried out as soon as possible to apply the circuit to the agricultural transport vehicle. At the same time, application research of permanent magnet generator for nuclear power generation is necessary.

(1) For design suitable for permanent magnet generator, mathematical model is built in synchronous rotation coordinate system, voltage-stabilizing principle is analyzed.

(2) In order to verify the rationality of the voltage-stabilizing principle, combining with simulation analysis, theoretical calculation, and experimental verification, it is confirmed that the voltage-stabilizing principle of the voltage-stabilizing circuit is correct, the selected component parameters are reasonable, and the simulation results are accurate and reliable.

(3) The voltage-stabilizing controller designed is installed on the permanent magnet generator and experimented on the automotive electrical experimental platform. The output voltage is between 13.82 V and 14.40 V, which conforms to the mechanical industry standard GB/T23903-2009 of the People's Republic of China. It is further proved that the voltage-stabilizing circuit has excellent voltage-stabilizing performance. By this way, the voltage-stabilizing circuit of permanent magnet generator for agricultural transport vehicles can be designed. It provides a convenient and reliable method for the design and manufacture of voltage-stabilizing circuit. The application of the voltage-stabilizing circuit greatly improves the performance of the electric system of the agricultural transport vehicles, reduces manufacturing cost and failure rate of the generator. The impact of COVID-19 pandemic and disruptions can be studied for transport vehicle energy management. In this regard, fuzzy, robust, and uncertain data can be predicted and optimized [37–40].

**Author Contributions:** Conceptualization, J.M. and L.S.; methodology, J.M.; software, J.M.; validation, J.M., L.S.; formal analysis, J.M.; investigation, Golmohammadi, A.-M.G.; resources, L.S.; data curation, L.S.; writing—original draft preparation, J.M.; writing—review and editing, A.-M.G.; visualization, L.S.; project administration, J.M.; funding acquisition, J.M. All authors have read and agreed to the published version of the manuscript.

**Funding:** This research was funded by National Natural Science Foundation of China, grant number 51975340; Xingtai Youth Talent Plan Project, grant number 2021ZZ034; Science and Technology Research Project of Colleges and Universities of Hebei Province, grant number QN2019205.

**Acknowledgments:** We thank Xueyi Zhang from Shandong university of technology for the support in the course of the experiment.

**Conflicts of Interest:** The authors declare no conflict of interest.

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
