# Peer review of "Voltage-Stabilizing Method of Permanent Magnet Generator for Agricultural Transport Vehicles"

_processes, doi:10.3390/pr10091726_

Round 1

Reviewer 1 Report

1.     Sentence incomplete in abstract. “In order to realize the stable output of permanent magnet generator”….

2.     English grammar need to be checked throughout the paper. “the voltage stabilizing circuit is design”..

3.     In the keyword, Permanent magnet; Generator is a single word

4.     Novelty and significant contribution by the authors should be elaborated at the end of introduction chapter.

5.     In figure 1, rotation speed unit is kr/min? Kindly check

6.     In page 6, line 198, ‘with’ is incomplete

7.     Many references are old one. Try to add few more recent references.

8.     Though experimental results have been carried out, the significant work and contribution coverage is missing in this paper. Improve this part.

Author Response

Dear Reviewer,

Based on your comments, we have responded.Please see the attachment!

Best regards,

All authors

Reviewer 2 Report

The paper contains all the necessary elements of the peer-review paper. The authors have comprehensive knowledge in the topic. The problem described in actual and needs further research. The applied mathematical and experiment set-up were designed with all international scientific rules and best practices.

I have just one serious concern about the paper. It was submitted to the Special Issue about nuclear energy. Just to be consistent with the Issue please add at the reference level some indication to the nuclear power, e.g. the voltage stabilization problem also can appear in the equipment of novel nuclear systems of IV Generation (e.g. High Temperature Reactors, https://doi.org/10.2478/nuka-2021-0020) and further research in this topic are necessary.

Author Response

(The authors gave the same response as above.)
